# Understanding Rad51 function is a prerequisite for progress in cancer research

Bengt Nordén[1] and Masayuki Takahashi[2]

[1]Chemistry and Chemical Engineering, Chalmers University of Technology, 412 96 Gothenburg, Sweden and [2]School of Bioscience and Biotechnology, Tokyo Institute of Technology, 2-12-1 Ookayama, Meguro-ku, Tokyo 152-8550, Japan

**Perspective**

**Key words:**
Rad51; homologous recombination; cancer; DNA repair; RecA; DNA strand exchange.

**Author for correspondence:**
Bengt Nordén, E-mail: norden@chalmers.se;
Masayuki Takahashi, E-mail: takahashi.m.ay@m.titech.ac.jp

## Abstract

The human protein Rad51 is double-edged in cancer contexts: on one hand, preventing tumour-igenesis by eliminating potentially carcinogenic DNA damage and, on the other, promoting tumours by introducing new mutations. Understanding mechanistic details of Rad51 in homologous recombination (HR) and repair could facilitate design of novel methods, including CRISPR, for Rad51-targeted cancer treatment. Despite extensive research, however, we do not yet understand the mechanism of HR in sufficient detail, partly due to complexity, a large number of Rad51 protein units being involved in the exchange of long DNA segments. Another reason for lack of understanding could be that current recognition models of DNA interactions focus only on hydrogen bond-directed base pair formation. A more complete model may need to include, for example, the kinetic effects of DNA base stacking and unstacking ('longitudinal breathing'). These might explain how Rad51 can recognize sequence identity of DNA over several bases long stretches with high accuracy, despite the fact that a single base mismatch could be tolerated if we consider only the hydrogen bond energy. We here propose that certain specific hydrophobic effects, recently discovered destabilizing stacking of nucleobases, may play a central role in this context for the function of Rad51.

## Causes of cancer

### Replication errors

Cancer is a genetic disease in the sense that it is induced by modification of genetic information (the nucleotide sequence) in genes involved in regulating cell proliferation resulting in uncontrolled cell proliferation, a landmark of tumour. Since cell proliferation is tightly regulated by both negative and positive controls, modification of more than two genes is usually required for tumour formation. Furthermore, cancer is a malignant tumour with potential of invasion into healthy tissue and metastasis. Modification of several other genes such as genes involved in apoptosis, metabolism, cell–cell contact, and so on, is also required for rapid proliferation, metastasis and escape from immune response (Fig. 1).

Modification could be caused by replication errors; this is, however, very rare because the high fidelity of DNA replication is maintained by DNA polymerase itself, its proofreading functions and other surveillance systems such as mismatch-repair systems (Kunkel, 2004; Bębenek and Graczyk, 2018). Therefore, modification of particular sets of several genes in one cell needs accumulation of errors upon many cell divisions over the course of a lifespan. Consequently, older people are more prone to get cancer. However, such events are still very rare under normal conditions. Loss of replication fidelity due to inherited and sporadic mutations in replication error repair systems (Bębenek and Graczyk, 2018) is a prerequisite for cancer formation. Individuals with inherited mutations in mismatch repair enzymes seem prone to developing colorectal cancer. Another cause that increases replication errors is DNA damage as we will touch upon below.

### DNA damage and error-prone repair

DNA damage by exogenous and endogenous agents can result in mutation (De Bont and van Larebeke, 2004; Tubbs and Nussenzweig, 2017). DNA is chemically fragile; every minute in every cell, it undergoes various chemical modifications, including oxidation, alkylation, deamination, elimination of nucleobases and disruption of the phosphate backbone (De Bont and van Larebeke, 2004). Some modifications, such as 8-oxoguanine (8-oxoG) and O-6 methyl guanine (O6-methylG), increase the likelihood of replication error by forming non-canonical base pairs (8-oxoG with A and O6-methylG with T). Other modifications are bypassed by less accurate DNA polymerases (translesion replication), promoting mutation (Sale, 2013).

Cells harbour multiple redundant DNA repair systems that help them avoid forming tumours (Sale, 2013). DNA damage that occurs on one of the two DNA strands can be repaired using the undamaged complementary strand as a template. By contrast, double-strand breaks cannot be fixed in this manner and are instead repaired by non-homologous end-joining (NHEJ),

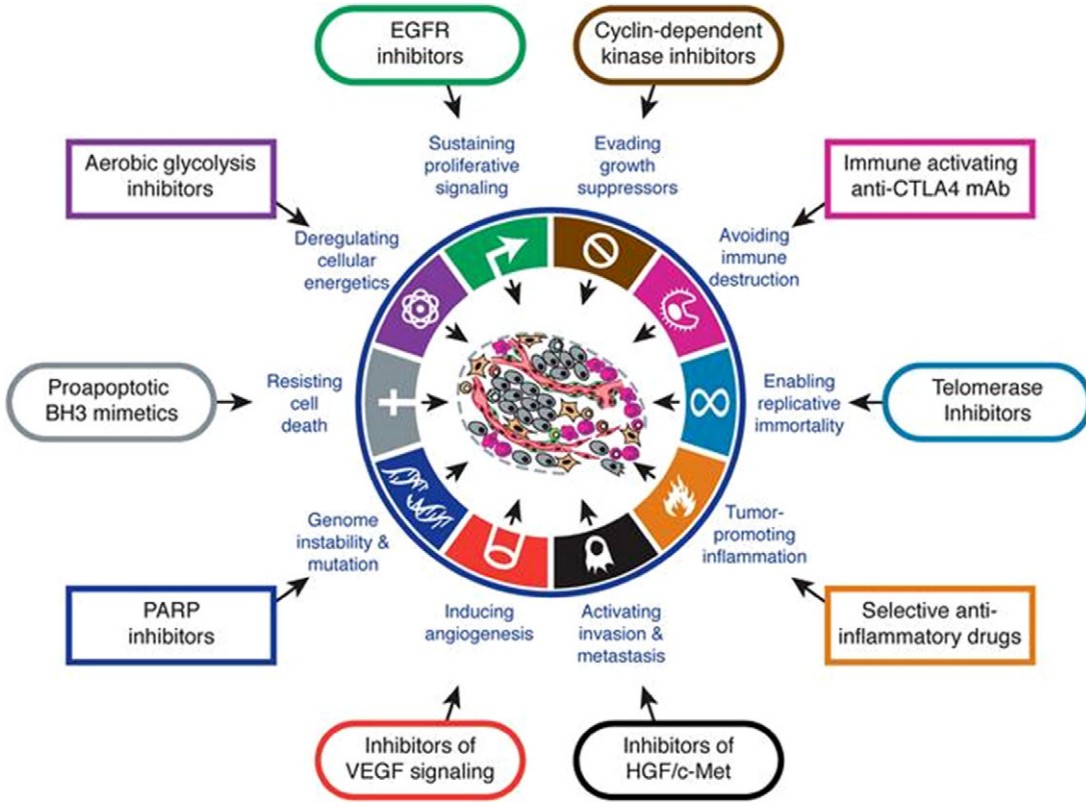

**Fig. 1.** Schematic view of available strategies for therapeutic targeting of the various 'hallmarks of cancer' with drugs that interfere with each of the acquired capabilities necessary for tumour growth and progression. Adapted with permission from Hanahan and Weinberg (2012).

microhomology-mediated end-joining or homologous recombination (HR) (Jasin and Rothstein, 2013; Wright *et al.*, 2018). NHEJ is mutagenic, whereas HR is comparatively error-free (Tubbs and Nussenzweig, 2017). Inherited and sporadic defects in repair enzymes predispose carriers to cancer formation, and such mutations are observed in many cancers (Tubbs and Nussenzweig, 2017). Behaviours that cause accumulation of DNA damage, such as smoking or excessive sun exposure, also increase cancer risk.

### HR: double-edged roles

HR involves exchange of strands between two homologous DNAs, and its main step is catalysed by Rad51 (Jasin and Rothstein, 2013; Wright *et al.*, 2018). A schematic presentation of the Rad51-promoted strand-exchange mechanism is given in Fig. 2. By exchanging a broken strand for an undamaged homologous strand, HR can repair double-strand breaks usually without causing any genetic modification and, in this way, prevent mutation. HR also restores a stalled replication fork due to replication error or non-treated DNA damage. However, the role of HR in preventing cancer was underestimated until the discovery that BRCA2, the mutation of which predisposes to breast and ovarian cancers with high penetrance (Prakash *et al.*, 2015; Heeke *et al.*, 2018), is involved in HR by direct interaction with Rad51 (Mizuta *et al.*, 1997).

On the other hand, strand exchange can promote loss of heterozygosity and aberrant genetic arrangement (Godin *et al.*, 2016). In our cells, most genes exist in pairs, one inherited from each parent. Mutations are rare and usually affect only one copy of a gene, and it is likely that any given mutation will be masked by the presence of the 'healthy' gene, with the exception of dominant phenotypes such as predisposition to breast cancer caused by

### Schematic presentation of Rad51-promoted DNA strand exchange reaction

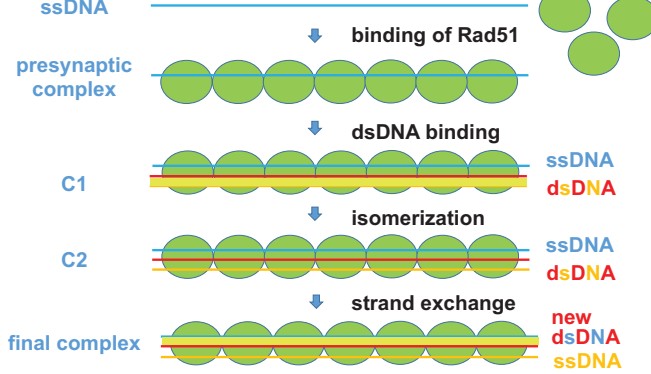

**Fig. 2.** Schematic presentation of Rad51-promoted DNA strand-exchange reaction. Reaction steps observed by kinetic analysis (Ito *et al.*, 2018) are shown. Rad51 molecules (green balls) bind to single-stranded DNA (blue line) to form a presynaptic filament. The filament binds a double-stranded DNA (red/orange lines) to form complex C1 and changes the DNA conformation (complex C1) before complete strand exchange. For simplicity, helical shapes of Rad51 filament and double-stranded DNA are not shown. Structures of C1 and C2 complexes are yet unknown.

*BRCA2* mutation. Loss of heterozygosity causes the mutant phenotype to become manifest. Thus, HR can in this way also contribute to cancer formation (Godin *et al.*, 2016). HR may also occur between two non-identical but similar DNA parts. This promotes deletion of DNA, when two parts are on the same DNA and chromosome translocation when two parts are on two different chromosomes (Bishop and Schiestl, 2002). These exchanges can occur in human

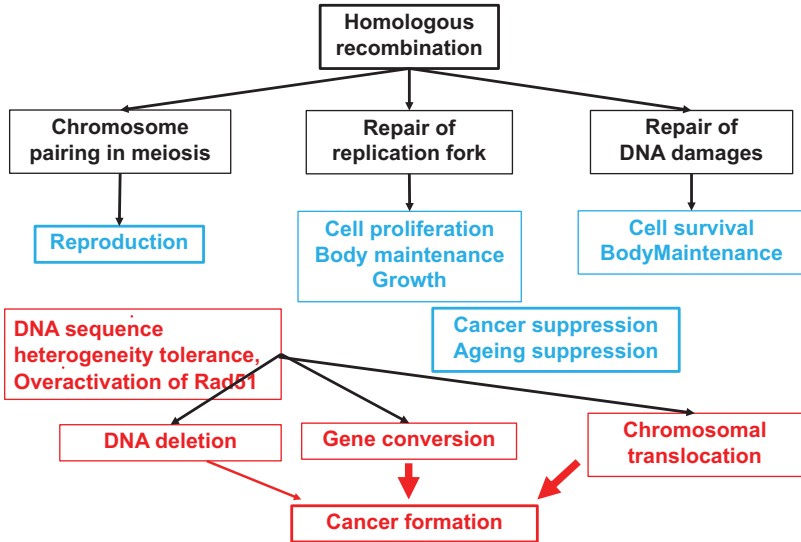

**Fig. 3.** The double-edged roles of Rad51-mediated homologous DNA recombination and associated normal, benign processes in an organism (blue) and below (in red) error-associated processes potentially connected to tumour formation.

because the human genome is so large that there can be many similar DNA sequences. Furthermore, the human genome contains many repetitive sequences. Therefore 'erroneous' HR may be a cause of cancer formation. In Fig. 3, the origin of the double-edged roles of Rad51 is schematically presented. Obviously, it is important to understand in detail the recognition mechanism of homologous DNA and its accuracy. Such knowledge would be also useful for the application of the CRISPR technique, which relies on accuracy of HR to incorporate external DNA at an appropriate site in chromosome.

## Rad51 and cancer

### Basic functions of Rad51

Rad51 is a member of the RecA family of proteins, which also includes RecA, RadA and Dmc1. It plays a key role in HR: searching out and pairing homologous DNA sequences and then promoting strand exchange between them (Fig. 2). Rad51 function in cells requires several accessory proteins (Bishop and Schiestl, 2002; Carreira and Kowalczykowski, 2011; Prakash *et al.*, 2015; Heeke *et al.*, 2018; Jia *et al.*, 2019), and its activity is regulated by the interactions with regulatory proteins and post-translational modifications. Rad51 is phosphorylated by cancer-related kinases (Daboussi *et al.*, 2002; Chabot *et al.*, 2019). Hypoxia in cancer cells is reported to downregulate Rad51 (Bindra *et al.*, 2004). In addition to defects in the accessory proteins of Rad51 involved in HR, which are frequently observed in cancer cells, alterations of Rad51 itself have also been observed in cancer patients (Antoniou *et al.*, 2008).

### Rad51 antagonises radio- and chemo-therapies of cancer

Rad51 is thought to protect cancer cells against radio- and chemo-therapies by repairing DNA damage induced by these treatments – 'Rad' derived from radiation (Prakash *et al.*, 2015). Consequently, cancers with defects in HR are more efficiently treated by such therapeutic modalities. In fact, inhibition or downregulation of Rad51 increases the efficiency of radiotherapy and chemotherapy (Tsai *et al.*, 2010). In this context, many small molecules that could act as inhibitors of Rad51 activity are worth investigating (Huang and Mazin, 2014; Chen *et al.*, 2017).

### Rad51 promotes cancer and cancer progression

When overexpressed, Rad51 can also promote cancer, and hyperactive Rad51 mutations have been detected in the cells of solid tumours. Multiple studies have reported Rad51 overexpression in various cancers (Godin *et al.*, 2016). Furthermore, these studies show that elevated Rad51 expression is correlated with reduced patient survival. Thus, Rad51 promotes progression of cancer, increasing its malignancy by stimulating metastasis and increasing resistance to chemotherapy. Accordingly, Rad51 is considered promising as a target for treatment of cervical carcinoma, breast cancer and non-small-cell lung cancer (Tsai *et al.*, 2010; Huang and Mazin, 2014; Chen *et al.*, 2017; Jia *et al.*, 2019).

Many cancers have epigenetic deficiencies in various DNA repair genes, likely resulting in higher rates of unrepaired DNA damage. The overexpression of Rad51 seen in many cancers may reflect compensatory Rad51 overexpression (as in *BRCA1* deficiency) and elevated rates of HR repair to deal at least partially with excess DNA damage.

## Challenge of understanding Rad51-promoted DNA strand-exchange reaction

### Appropriate inhibition of Rad51

As noted above, Rad51 has been proposed as a target for cancer treatment. Indeed, some Rad51 inhibitors are effective against some cancers. However, because of the double-edged character of Rad51, inhibition of its activity may also have negative effects, for example, promoting more malignant cancers. Many currently available Rad51 inhibitors prevent formation of Rad51 filament on DNA, the first step of the reaction (Huang and Mazin, 2014; Chen *et al.*, 2017). Consequently, damage then remains unaddressed and may end up being processed by a mutagenic repair system such as NHEJ. Therefore, it would be preferable to inhibit the reaction at an intermediate stage that is toxic to cancer cells. Such toxic Rad51 filaments have been reported in yeast (Veaute *et al.*, 2003; Symington and Heyer, 2006). However, it is necessary to fully understand the reaction mechanism in order to inhibit Rad51 activity at the appropriate step to avoid negative effects.

### Rad51 will be crucially needed in future CRISPR-based cancer therapy

Double-strand-break repair by HR is initiated by 5′-to-3′ strand resection; in humans, the DNA nuclease cuts back the 5′ end of one strand to generate a 3′ single-stranded DNA overhang. The development of CRISPR technology for both cancer gene diagnostics and therapeutics (including gene-edited T-cells) will rely on insight into the native cellular repair system. This may represent one of the most important motivations for investigating Rad51 (and RecA, as model system) in cancer research. When Aaron Klug invented zinc finger-based sequence-specific artificial double-strand endonucleases, he noted that a bottleneck in progress toward gene-correction therapy would be the need to entrust insertion of the desired DNA into the native cellular recombination machinery, that is, Rad51 (Deltcheva et al., 2011; Jinek et al., 2012).

### Reaction mechanism

The mechanisms of recombination enzymes appear similar in all organisms (Ito et al., 2018; Takahashi, 2018). In eukaryotes, the Rad51 protein plays a central role in homologous recombinational repair. Specifically, Rad51 catalyses strand transfer between a damaged sequence and its undamaged homologue to allow re-synthesis of the damaged region. In addition, it collaborates with Dmc1, another recombinase, in proper segregation of chromosomes in meiosis. For the reaction, the enzyme binds to a single-stranded region of DNA to form a well-organized filamentous complex with DNA (Fig. 2). This single-stranded (ss) DNA–RecA filament then interacts with a double-stranded (ds) DNA to form an ssDNA–RecA–dsDNA complex, and if the two DNAs have identical or nearly identical sequences, strand exchange occurs. All reaction steps occur in a long nucleoprotein filament.

The structure of ssDNA–RecA filament has been determined by X-ray crystallographic analysis (Chen et al., 2008) and other techniques such as linear dichroism and electron microscopy (Stasiak et al., 1981; Norden et al., 1992). The structure of ssDNA–Rad51 filament seems similar (Xu et al., 2017). In the filament, Rad51 monomers are arranged in a helical manner around ssDNA, which also forms helix and ready to receive dsDNA (Reymer et al., 2009). DNA is elongated about 50% in length. By contrast, the structure of ssDNA–RecA–dsDNA (and ssDNA–Rad51–dsDNA) complex has not been determined although linear dichroism and modeling indicate a similar protein arrangement (Reymer et al., 2009). Kinetic analysis shows the presence of two ssDNA–Rad51–dsDNA complex intermediates (Fig. 2) (Ito et al., 2018). Recognition of sequence homology seems to occur before formation of Watson–Crick base pairing between ssDNA and complementary strand separated from dsDNA (Gupta et al., 1999). The reaction can start at any part with six-base homology (Anand et al., 2017; Takahashi, 2018).

Despite the great importance of recombinases in the context of human health (e.g. cancer, gene therapy and sterility) and many years of intense research, the mechanisms of homology search and strand exchange are not yet understood at the atomic level. Many questions, including how the Rad51 filament binds a second DNA, recognizes and searches sequence homology and why Rad51 stretches the DNA, remain unanswered. An improved understanding of the mechanistic details of these fundamental processes could hopefully clarify the accuracy of HR and its double-edged roles in both cancer prevention and formation. Such knowledge will also be useful for the CRISPR technology, in which incorporation of new DNA relies on the cell's native recombination machinery.

The lack of breakthroughs in HR research could have several explanations. One possibility could be the fact that the reaction differs from many other enzymatic reactions in terms of substrate size. Rad51 exchanges strands of long DNA segments, and many Rad51 molecules must assemble into a very long filament to catalyse the reaction. Consequently, elucidating reaction mechanisms is challenging. Furthermore, Rad51 interacts with two DNA molecules to promote strand exchange between them. The details of how Rad51 interacts with these DNAs, especially the second one, and associated topological challenges, remain elusive. For example, we do not know how the second DNA enters the filament to interact with the first DNA, which is completely coated by Rad51. Similarly, it remains unclear how Rad51 separates the two strands of double-stranded DNA to promote strand exchange.

### Contributions of other factors than hydrogen bonds to DNA base pair recognition

A provocative possibility is that our current model of DNA recognition based on hydrogen bonded Watson–Crick base pairs is somehow incomplete, requiring some new theoretical approach to guide more systematic experiments. Here key factors including hydrophobic catalytic effects may have been overlooked. Conventionally, we imagine that the HR in DNA involves the formation of base pairs mediated by hydrogen bonds between two strands. However, hydrogen bonding alone cannot explain the discrimination of DNA with few mismatches relative to a completely complementary strand. Both Rad51 and RecA initiate the HR reaction when the two DNAs present sequence identity over as many as six to eight contiguous bases (Anand et al., 2017; Takahashi, 2018). The energetic difference between binding of DNA with one mismatch and with a completely matched DNA is too small (less than 20%) if we consider only hydrogen bond formation between bases. Still RecA manages to eliminate such a singly mismatched DNA in more than 80% of the cases (Takahashi, 2018). This may be because the selection is not made thermodynamically but kinetically (Bazemore et al., 1997; Takahashi, 2018). A matched DNA binds to the Rad51/DNA filament much faster than one containing a mismatch, as has also been observed in hybridization contexts (Jensen et al., 1997). This is apparently also the case with the selection of complementary base by DNA polymerase during replication (Oertell et al., 2016). However, the hydrogen bonds become rapidly weaker with distance, and it is difficult to imagine that hydrogen bonds alone could play a decisive role in the kinetic selection process. We here propose that other factors be involved.

We know today that DNA polymerase selects the correct DNA base without requiring complete hydrogen bond formation (Kool, 2000), and that certain modified DNA bases that cannot form hydrogen bonds with a partner are still selected by polymerases. Moreover, several studies have reported artificial base pairs involving non-canonical bases that do not form any hydrogen bonds with their partners (Henry and Romesberg, 2003; Yamashige et al., 2012), and that DNA containing such a third base pair (in addition to A–T and G–C pairs) can still be correctly replicated without significant error. In these base pairs, geometrical complementarity of two bases appear to be important (Yamashige et al., 2012). For such a recognition mechanism, also the correct mutual orientation of the two bases is crucial. This orientation effect may be the result of a precise steric guidance due to base stacking with adjacent bases. A similar mechanism, we propose, may be involved in the Rad51-promoted DNA recognition and strand exchange.

The stability of double-stranded DNA is supported mainly by base stacking between neighbouring bases, rather than by formation of A–T and G–C base pairs (Yakovchuk *et al.*, 2006). We recently demonstrated that the semi-hydrophobic environment created by ethylene glycol ethers, such as poly-ethylene glycol (PEG), significantly weakens DNA base-stacking strength but not the hydrogen bonding, and also promotes the DNA strand-exchange reaction even in the absence of any recombinase protein (Feng *et al.*, 2019). Interestingly, both RecA and Rad51 elongate DNA by unstacking one base pair of every three, evidencing a weakening of the base-stacking forces (Chen *et al.*, 2008; Xu *et al.*, 2017; Sun *et al.* 2020). Interestingly, we also note that several hydrophobic residues are present in the proximity of the DNA-binding sites of both Rad51 and RecA (the so-called L1 and L2 loops), and, thus, we speculate that these, just like PEG, can provide a hydrophobic environment that may catalyse the strand-exchange reaction.

## Perspective

Obviously, Rad51 is involved in causing many, possibly a majority, of our common cancers. At present, however, this is simply a statement of correlation, and the exact causal mechanism is yet unknown. It is obvious that dealing with the genome, either by natural heterozygotic gene mixing, which is the basis of Darwinian evolution, or the repair of random errors is serious business and prone to introduce potentially malignant genetic aberrations. Unsolved questions regarding both the eukaryotic Rad51 and its simpler, more 'ancestral' protein RecA in prokaryotes have persisted despite over 40 years of intense research. This may be partly explained by the inherent complexity of the HR reaction, which involves a large number (10–100) of Rad51 protein units bound to a stretch of DNA. However, it is also possible that our understanding of recombination mechanisms remains incomplete. One factor recently proposed to be important for the interactions and stability of DNA is hydrophobic effects in the vicinity of DNA, which may be involved in catalysing crucial steps in the HR reaction (Ito *et al.*, 2018) via a destabilized base stacking (Feng *et al.*, 2019). Such an effect might explain the bewildering observation that in contrast to DNA replication and transcription, where nucleotides are added and tested one by one, homologous DNA recombination through strand exchange is testing whole sequences, six or more bases in length, for complementarity, requiring the elimination of partially mismatched yet thermodynamically stable intermediates. These are just a few of the elusive points; many questions remain regarding mechanism of gene recombination. For these reasons, and in light of the strong connection between cancer and Rad51, we conclude that fundamental research on Rad51 is a prerequisite for progress in cancer research.

**Open Peer Review.** To view the open peer review materials for this article, please visit http://doi.org/10.1017/qrd.2020.13.

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
