## [Reviewer Report]

*Comments to Author*: Nordén & Takahashi write a Perspective with a thought-provoking title: “Understanding Rad51 function is a prerequisite for progress in cancer research”. Cancer is an extremely complex molecular and cellular phenomenon.From their own point of view, they summarized how Rad51 plays a double-edged role: 1) positive role to protect further DNA damage, repairing it through homologue recombination preventing cancer development, 2) negative role to prevent chemotherapy treatment and participating cancer further development.

In the Abstract and a variation of it later in the Conclusion, they state that “Another, more dramatic reason for lack of understanding could be that our model of DNA interaction based on hydrogen bond-directed base pair formation is somehow fundamentally wrong, mandating a complete overhaul of the way we think of reactions involving DNA. Since Rad51 recognizes sequence identity of DNA over several bases, one base mismatch could be tolerated when we consider only hydrogen bond energy. This is not the case. We propose that certain hydrophobic effects, recently discovered destabilizing stacking of nucleobases may play a central role in this context.”

This reviewer considers their statement very provocative and thought-provoking.But the statement is too strong to be readily accepted by the research community.It is better to state that the current understanding and emphasis of importance of hydrogen bond for conferring DNA stability is incomplete. The accumulated experimental data requires new theoretical insight to guide experiments into how hydrophobic interaction also plays a key role for DNA stability and in homologue recombination.

The importance of hydrogen bond in DNA base pairing is indisputable: i) faithful DNA replication with extremely rare errors, ii) RNA transcription from DNA with very high fidelity and iii) the genetic code for specific tRNA anticodon to recognize the codon mRNA are all require specific hydrogen bond recognition.The base stacking through the hydrophobic interaction in DNA is absolutely critical for DNA stability and hybridization, but hydrophobic interaction cannot confer specificity of base pairing.It is known that DNA hybridization is largely driven by hydrophobic interaction in water, non-aqueous solvent can interfere the hybridization both thermodynamically and kinetically. But for specific base pairing, namely, how each base recognizes its complementary base requires precise hydrogen bond recognition and perfect registration. The author is correct that hydrogen bonds contribute less to the DNA stability and is not the driving force for hybridization.The hydrogen bonds main function is to confer and confirm the base-pairing specificity, which is the essence of high fidelity of replication, transcription and translation.

Rad51 is a very interesting protein and plays many roles in cell. But its negative tole to promote cancer was less widely known. They wrote “When overexpressed Rad51 can also promote cancer, and hyperactive Rad51 mutations have been detected in the cells of solid tumors. Multiple studies have reported Rad51 overexpression in various cancers (Godin et al., 2016). Furthermore, these studies show that elevated Rad51expression is correlated with reduced patient survival.” In light of the latest clinical observations, paying closer attention and intense scrutiny to Rad51’s other role I cancer is required.

In order to truly understand the double-edged role of Rad51, additional experiments are needed including the detailed molecular structure with DNA in action.The latest CryoEM may be a useful tool to address such dauting structural question.But it will likely gain much insights.

A few points to consider:

1) In order to clearly emphasis of the double-edged role of Rad51, this reviewer suggests to add a Figure 3. This figure 3 can be a cartoon to clearly show Rad51 dual function: i) an important positive role in DNA homologue recombination, ii) negative role in participating cancer development, e.g. phosphorylated by cancer-related kinases.

2) On page 5, 2nd paragraph, line 8, “recombination may also occur between two no identical but similar DNA parts”, it is better to use “between two nonidentical but similar DNA parts”.

3) On page 10, 2nd paragraph, it is best to add a subtitle for this section to emphasize author’s points.

4) The statement “A third, more dramatic possibility is that something is wrong with our basic theory of DNA interactions, which may require a complete paradigm overhaul” is too strong.It could be re-written as “A more provocative possibility is that our current knowledge on DNA interaction strongly emphasis on hydrogen bond is incomplete, new theoretical insights to guide more systematic experiments are required. Other key factors including hydrophobic catalysis should not be overlooked.”

5) Page 10, the last paragraph, they state: “Another possibility is that hydrogen bond formation is less important, and that some other element governs selection”.A phrase could be inserted after less important “it may not be the driving force for DNA hybridization and double helix stability”.

6) Page 11, 2nd paragraph, “We must remember that the stability of double-stranded DNA”, “We must remember that” is unnecessary.

7) Page 12, the subtitle “Conclusion”.It is better to use “Perspective”.

8) First line, “Obviously, Rad51 is somehow involved”. It is not better not to use the word “somehow”.Somehow is not science.Science must explain “How”, not “Somehow”.If it is uncertain, the phrase could change to “Rad51 may be involved”.

9) Page 12, line 6, “Mysteries”, it is better to use “Unsolved questions”.

10) Line 9, “that our understanding of recombination mechanisms remains incomplete because something is fundamentally wrong with our basic theory of DNA interactions.” The sentence “because something is fundamentally wrong with our basic theory of DNA interactions” is not necessary since you immediately below proposed your own idea to complete the incomplete theory.

After the authors add Figure 3 to clearly show the double-edged role of Rad51, address these concerns and make careful revision, this reviewer recommends publication of this provocative and thought-provoking Perspective.

---

## [Reviewer Report]

*Comments to Editor*: This is a very interesting short review that fits well to QRB-Discovery. My minorsuggestions are indicated below.’

*Comments to Author*: Abstract: line 10: “fundamentally wrong” seems too harsh.

On p. 10 the paragraph “A third, more dramatic…” could be expanded to some extent. I think that at least some of the readers know that it is not only the H bonds between the bases that stabilize the double helix. It could here be interesting to see more data ( from the references?) about the theoretical calculations that describe the energetics of the double helix.

---

## [Reviewer Report]

*Comments to Editor*: No accompanying comment.

*Comments to Author*: Reviewer #1: Nordén & Takahashi write a Perspective with a thought-provoking title: “Understanding Rad51 function is a prerequisite for progress in cancer research”. Cancer is an extremely complex molecular and cellular phenomenon.From their own point of view, they summarized how Rad51 plays a double-edged role: 1) positive role to protect further DNA damage, repairing it through homologue recombination preventing cancer development, 2) negative role to prevent chemotherapy treatment and participating cancer further development.

In the Abstract and a variation of it later in the Conclusion, they state that “Another, more dramatic reason for lack of understanding could be that our model of DNA interaction based on hydrogen bond-directed base pair formation is somehow fundamentally wrong, mandating a complete overhaul of the way we think of reactions involving DNA. Since Rad51 recognizes sequence identity of DNA over several bases, one base mismatch could be tolerated when we consider only hydrogen bond energy. This is not the case. We propose that certain hydrophobic effects, recently discovered destabilizing stacking of nucleobases may play a central role in this context.”

This reviewer considers their statement very provocative and thought-provoking.But the statement is too strong to be readily accepted by the research community.It is better to state that the current understanding and emphasis of importance of hydrogen bond for conferring DNA stability is incomplete. The accumulated experimental data requires new theoretical insight to guide experiments into how hydrophobic interaction also plays a key role for DNA stability and in homologue recombination.

The importance of hydrogen bond in DNA base pairing is indisputable: i) faithful DNA replication with extremely rare errors, ii) RNA transcription from DNA with very high fidelity and iii) the genetic code for specific tRNA anticodon to recognize the codon mRNA are all require specific hydrogen bond recognition.The base stacking through the hydrophobic interaction in DNA is absolutely critical for DNA stability and hybridization, but hydrophobic interaction cannot confer specificity of base pairing.It is known that DNA hybridization is largely driven by hydrophobic interaction in water, non-aqueous solvent can interfere the hybridization both thermodynamically and kinetically. But for specific base pairing, namely, how each base recognizes its complementary base requires precise hydrogen bond recognition and perfect registration. The author is correct that hydrogen bonds contribute less to the DNA stability and is not the driving force for hybridization.The hydrogen bonds main function is to confer and confirm the base-pairing specificity, which is the essence of high fidelity of replication, transcription and translation.

Rad51 is a very interesting protein and plays many roles in cell. But its negative tole to promote cancer was less widely known. They wrote “When overexpressed Rad51 can also promote cancer, and hyperactive Rad51 mutations have been detected in the cells of solid tumors. Multiple studies have reported Rad51 overexpression in various cancers (Godin et al., 2016). Furthermore, these studies show that elevated Rad51expression is correlated with reduced patient survival.” In light of the latest clinical observations, paying closer attention and intense scrutiny to Rad51's other role I cancer is required.

In order to truly understand the double-edged role of Rad51, additional experiments are needed including the detailed molecular structure with DNA in action.The latest CryoEM may be a useful tool to address such dauting structural question.But it will likely gain much insights.

A few points to consider:

1) In order to clearly emphasis of the double-edged role of Rad51, this reviewer suggests to add a Figure 3. This figure 3 can be a cartoon to clearly show Rad51 dual function: i) an important positive role in DNA homologue recombination, ii) negative role in participating cancer development, e.g. phosphorylated by cancer-related kinases.

2) On page 5, 2nd paragraph, line 8, “recombination may also occur between two no identical but similar DNA parts”, it is better to use “between two nonidentical but similar DNA parts”.

3) On page 10, 2nd paragraph, it is best to add a subtitle for this section to emphasize author’s points.

4) The statement “A third, more dramatic possibility is that something is wrong with our basic theory of DNA interactions, which may require a complete paradigm overhaul” is too strong.It could be re-written as “A more provocative possibility is that our current knowledge on DNA interaction strongly emphasis on hydrogen bond is incomplete, new theoretical insights to guide more systematic experiments are required. Other key factors including hydrophobic catalysis should not be overlooked.”

5) Page 10, the last paragraph, they state: “Another possibility is that hydrogen bond formation is less important, and that some other element governs selection”.A phrase could be inserted after less important “it may not be the driving force for DNA hybridization and double helix stability”.

6) Page 11, 2nd paragraph, “We must remember that the stability of double-stranded DNA”, “We must remember that” is unnecessary.

7) Page 12, the subtitle “Conclusion”.It is better to use “Perspective”.

8) First line, “Obviously, Rad51 is somehow involved”. It is not better not to use the word “somehow”.Somehow is not science.Science must explain “How”, not “Somehow”.If it is uncertain, the phrase could change to “Rad51 may be involved”.

9) Page 12, line 6, “Mysteries”, it is better to use “Unsolved questions”.

10)Line 9, “that our understanding of recombination mechanisms remains incomplete because something is fundamentally wrong with our basic theory of DNA interactions.” The sentence “because something is fundamentally wrong with our basic theory of DNA interactions” is not necessary since you immediately below proposed your own idea to complete the incomplete theory.

After the authors add Figure 3 to clearly show the double-edged role of Rad51, address these concerns and make careful revision, this reviewer recommends publication of this provocative and thought-provoking Perspective.

Reviewer #2: Abstract: line 10: “fundamentally wrong” seems too harsh.

On p. 10 the paragraph “A third, more dramatic…” could be expanded to some extent. I think that at least some of the readers know that it is not only the H bonds between the bases that stabilize the double helix. It could here be interesting to see more data ( from the references?) about the theoretical calculations that describe the energetics of the double helix.

Please foloww tehereferees sugestions